# Detection of Cigarette Smoke Using a Surface-Acoustic-Wave Gas Sensor with Non-Polymer-Based Oxidized Hollow Mesoporous Carbon Nanospheres

**DOI:** 10.3390/mi10040276

**Published:** 2019-04-24

**Authors:** Chi-Yung Cheng, Shih-Shien Huang, Chia-Min Yang, Kea-Tiong Tang, Da-Jeng Yao

**Affiliations:** 1Institute of NanoEngineering and MicroSystems, National Tsing Hua University, Hsinchu 30013, Taiwan; marybrian5498@gmail.com; 2Department of Chemistry, National Tsing Hua University, Hsinchu 30013, Taiwan; sean00132032@gmail.com (S.-S.H.); cmyang@mx.nthu.edu.tw (C.-M.Y.); 3Department of Electrical Engineering, National Tsing Hua University, Hsinchu 30013, Taiwan; kttang@ee.nthu.edu.tw; 4Department of Power Mechanical Engineering, National Tsing Hua University, Hsinchu 30013, Taiwan

**Keywords:** surface acoustic wave, second-hand smoke, 3-ethenylpyridine, oxidized hollow mesoporous carbon nanosphere

## Abstract

The objective of this research was to develop a surface-acoustic-wave (SAW) sensor of cigarette smoke to prevent tobacco hazards and to detect cigarette smoke in real time through the adsorption of an ambient tobacco marker. The SAW sensor was coated with oxidized hollow mesoporous carbon nanospheres (O-HMC) as a sensing material of a new type, which replaced a polymer. O-HMC were fabricated using nitric acid to form carboxyl groups on carbon frameworks. The modified conditions of O-HMC were analyzed with Scanning Electron Microscopy (SEM), Fourier transform infrared spectrometry (FTIR), and X-ray diffraction (XRD). The appropriately modified O-HMC are more sensitive than polyacrylic acid and hollow mesoporous carbon nanospheres (PAA-HMC), which is proven by normalization. This increases the sensitivity of a standard tobacco marker (3-ethenylpyridine, 3-EP) from 37.8 to 51.2 Hz/ppm and prevents the drawbacks of a polymer-based sensing material. On filtering particles above 1 μm and using tar to prevent tar adhesion, the SAW sensor detects cigarette smoke with sufficient sensitivity and satisfactory repeatability. Tests, showing satisfactory selectivity to the cigarette smoke marker (3-EP) with interfering gases CH_4_, CO, and CO_2_, show that CO and CO_2_ have a negligible role during the detection of cigarette smoke.

## 1. Introduction

Second-hand smoke (SHS), which is also called environmental tobacco smoke (ETS) or passive smoking, is a serious environmental pollution, composed of the main-stream smoke exhaled by active smokers and the side-stream smoke expelled from the lit tobacco product, such as a cigarette [1]. More than 4700 substances are now recognized as constituents of cigarette smoke [2]. More than 60 compounds are known to cause cancer in a human body [3], such as tar (including mutagenic and carcinogenic agents produced from incomplete combustion during smoking), tobacco-specific nitrosamines (TSNA), polonium-210, polycyclic aromatic hydrocarbons (PAH), etc. [4]. SHS is harmful for everybody, especially pregnant women and children. Statistical data reveal that people exposed to SHS have a greater risk of lung cancer (20–30%), oral cancer, asthma, chronic obstructive pulmonary disease (COPD), coronary heart disease (25%), and childhood illness [5,6]. A medical report states that, from 2010 to 2014 in the USA, smoking rates of adults fell from 43% (1965) to 18% today, but the number of deaths caused by smoking and exposure to SHS is still estimated to be about 480,000 per year [7].

In the complicated mixture of SHS, nicotine and 3-ethenylpyridine (3-vinylpyridine, 3-EP) are two important markers for sensing cigarette smoke because they are specific to tobacco and the vapor phase in SHS. 3-EP is a pyrolysis product of nicotine [8]. Even though the concentration of 3-EP in a smoke area is typically less than nicotine, the stable characteristics and smaller rate of surface absorption of 3-EP leads to it still being used as an SHS marker in various related research [9]. Most measurements of ambient nicotine are analyzed with a gas chromatograph - mass spectrometer (GC-MS) and typically have a prolonged sampling period of days because of the small effective rates of sampling [10,11]. At present, conductive polymer-based sensors have been designed to monitor a tobacco-specific vapor such as nicotine or 3-EP for real-time detection, but some drawbacks such as a lack of repeatability or a long response time must be improved [12,13].

Devices based on microelectromechanical systems (MEMS) have been widely used in many areas such as particle separation [14], chemical detection [15,16], biomedical detection [17], and microfluidics [18,19]. Surface-acoustic-wave (SAW) devices are easily manufactured with MEMS techniques and are used for gas sensing. The basic principle of a SAW sensor is that the coated interdigital transducer (IDT) stimulates a piezoelectric substrate with a stable electrical resonant frequency when a radio frequency (RF) signals in and converts it to propagate mechanical waves along the surface of the device, which is known as a surface acoustic wave. When the target is adsorbed by the sensing layer, the surface acoustic wave slows because of the mass loading. The resonant frequency decreases and the mass change would be proportional to the gas concentration observed as a frequency shift. Because the energy of SAW highly concentrates on the substrate surface [20], it is sensitive to any perturbation on the surface such as temperature, mass change, conductivity, and elasticity. These properties endow SAW devices with many advantages such as extremely high sensitivity, small size, a wireless sensing platform, and a low working temperature [15].

The sensitive coating of an SAW sensor is typically a polymer film with a suitable functional group, polarity, molecular geometry, and more. Some devices in the literature combine nanomaterials such as nanowire, carbon nanotube (CNT), graphene into the polymer for improved detection performance on increasing the sensing area, and leading to an increased mass loading [21,22,23]. However, a polymer-based sensing film has some drawbacks including lack of thermal stability and a mismatch between glass transition temperature and ambient temperature. Furthermore, some target vapor permeates into the polymer causing the polymer film to expand, called a swelling effect [24]. In this research, a surface-acoustic-wave (SAW) sensor is used to detect an SHS marker at a small concentration and cigarette smoke with sensing materials of two kinds – hollow mesoporous carbon nanospheres (HMC) combined with a polymer as a nanocomposite, and a modified nanomaterial without a polymer called oxidized hollow mesoporous carbon nanospheres (O-HMC). A sensitive and real-time cigarette sensor can help one avoid the SHS damage in public places such as a hospital and a school.

## 2. Materials and Methods

### 2.1. Preparation of a Surface-Acoustic-Wave (SAW) Sensor

The SAW chips with a delay line were fabricated with MEMS techniques on a 128° YX-lithium niobate (LiNbO3) piezoelectric substrate, which has a large electromechanical coupling coefficient (K2, 5.5%) and large velocity (3992 m/s) [25]. The delay line consisted of a pair of input and output interdigital transducers (IDT) with each having 50 pairs of fingers with gold (thickness 100 nm) and chromium (20 nm) deposited with an e-gun evaporator. The acoustic wavelength of the interdigital transducers (IDT) is designed to be about 34 μm. The operating frequency of the SAW device is about 114.2 MHz near 23 °C The insertion losses (IL) range from 6 to 9 dB, which is characterized with a network analyzer.

To complete the fabrication, the finished wafer was cut into chips. The dimensions of each device were 13.4 mm × 7.4 mm × 0.5 mm. To decrease the gas diffusion volume and to decrease the response time, a self-designed micro-chamber (800 μL) is mounted on a printed-circuit board (PCB). The gas is delivered with dispensing needles, which is shown in Figure 1a.

### 2.2. Sensitive Materials

In this research, sensing materials of two types were chosen for the detection of cigarette smoke. One is a polymer-based and the other is only a nanostructure, without a polymer. For a polymer-based sensing film, poly-acrylic acid (PAA) is used for its universality. It has a glassy state near 23 °C because its glass transition temperature is about 106 °C, so it is hard to adsorb a target like a flexible polymer film [26]. To increase the reaction area and to decrease the response time, hollow mesoporous carbon nanospheres (HMC) are used to produce a nanocomposite sensing material (PAA-HMC). Each HMC has a specific surface area about 2350 m^2^/g. The average size is in the range of 80 to 120 nm with a porosity of 4 nm.

To avoid the disadvantages of a polymer, a non-polymer-based sensing material, oxidized hollow mesoporous carbon nanospheres (O-HMC) is a suitable material for detection since it combines the properties of functional groups of a polymer (carboxyl group) and a large specific surface area of a nanomaterial. The fabrication of O-HMC is simple, modified by liquid-phase oxidation of HMC directly, and treated with nitric acid (2.5 M) at an appropriate temperature and for an appropriate period to introduce functional groups on the carbon framework [27]. These two sensing materials are solid after fabrication. In attempt to facilitate the coating, a sensing material (10 mg) is added to ethanol (1 mL) as a solvent. Before coating, the aggregation of this sensing material was prevented through ultra-sonication for 20 min, spin coating with great uniformity at 1500 rpm, and heating at 90 °C for 10 min to remove the solvent, which is shown in Figure 1b. The black dots on the delay line are the O-HMC.

### 2.3. Sensing System

The SAW sensing system is shown in Figure 1c. A standard gaseous SHS marker (3-EP) is prepared on dipping 3-ethenylpyridine liquid into a polytetrafluoroethene-based (PTFE) sampling bag to produce a saturated vapor. A three-way valve installed on the sampling bag can switch a sample or dry air from an air compressor (relative humidity about 20%). A sample of cigarette smoke is collected from a lit cigarette (Marlboro, tar 10 mg and nicotine 0.8 mg per cigarette) and placed on the bottom of a glass syringe. The cigarette smoke was drawn with a pump (Thomas, diaphragm pump 2002, 400 mL/min) into a sampling bag. Before detection of the cigarette smoke, the sampling bag was left at ambient temperature for 15 min to preclude any influence of temperature. For a dynamic detection, a gas sample was drawn into the micro-chamber with a peristaltic pump (flow rate of 20 mL/min). At the beginning of tests, dry air flowed into the chamber until the frequency response was stable. The gas sample channel was then switched for adsorption, and, eventually, the dry air valve was opened again to purge the sensing material and to complete one cycle. Since the temperature and humility effects by a piezoelectric subtract would be relatively large, the un-coated sensing chip would be used to eliminate the environmental interferences. All detection took an untreated HMC-coated SAW sensor as a reference for improved stability [15,16,28,29]. The frequency response was recorded in a computer with a general purpose interface bus (GPIB) card, which was connected to a frequency counter, and which monitored the frequency change between the sensing chip and the reference.

## 3. Results

### 3.1. Sensitive Analysis of a Material

The surface morphologies of HMC and O-HMC with varied chemically modified parameters were analyzed using a scanning electron microscope (SEM), as shown in Figure 2. The original HMC showed the size to range from 80 to 120 nm and its mesopores to have a diameter about 4 nm, which is shown in Figure 2a. After oxidation, some carbon species were cleaved from the carbon nanospheres, which made an incomplete spherical appearance shown in Figure 2b. When the modified conditions are too harsh, the carbon nanospheres are damaged with HNO_3_, which leads to structural collapse to decrease the pore size. This might be explained by the repair of surface oxygen on the carbon walls [27]. The O-HMC seem to crosslink with each other, shown in Figure 2c, and show a serious aggregation problem that many O-HMC aggregate into clumps of diameter greater than 1 μm, which is shown in Figure 2d.

Considering the sensing application, the chemical structure of HMC and O-HMC was characterized with infrared spectra (FTIR) and scanned in a range from 500 to 4000 cm^−1^, as shown in Figure 3. The HMC before and after oxidation seem to have a similar spectrum, but show a significant difference of the C=O vibrational line at 1730 cm^−1^. A comparison with the spectrum before oxidation indicates that this signal denotes the presence of carbonyl or carboxyl groups. When HMC were treated with nitric acid, the stronger oxidation increased the intensity at 1730 cm^−1^, which means that more carboxyl groups were modified.

The XRD pattern of the HMC shows three signals that are indexed as (110), (211), and (220) reflections of structure Ia3d, shown as Figure 4 [30]. A transformation of structure occurred after the removal of the silica template. The carbon framework became atomically disordered, as revealed by the (110) signal appearing. The (110) reflection is symmetrically forbidden for Ia3d [31]. Too much oxidation leads to structural changes, revealed by the weakened (110) and (211) XRD reflections by damaging the original structure or cleavage of the carbon. Based on the above analysis, to retain sufficient modification for great sensitivity and the complete nanostructures for a large surface area, O-HMC-80 °C-15 h was chosen as the appropriate sensing material for subsequent detection.

### 3.2. Detection of the Standard Cigarette Marker

Figure 5 shows the frequency response of 3-EP detection (3 ppm) with sensing material of three types. It clearly shows that the sensing layer of O-HMC without a polymer is more sensitive than polymer-based PAA-HMC for detection of 3-EP, which is a greater frequency shift of 90 Hz. Taking 60 s for the adsorption, the desorption time is 90 s. Hydrogen bonding between the carboxylic acid and 3-EP become desorbed to the original frequency. Showing an effective reversible detection, this result proves that the modified O-HMC is a suitable choice to replace a polymer-based sensing material (PAA-HMC). In addition, the non-treated HMC shows slight adsorption, likely because some 3-EP is trapped in the porous nanostructure.

### 3.3. Normalization

The mass loading of the sensitive layer is an important factor for the sensor performance. In general, more sensing material coating can cause a greater frequency shift. However, too much sensing material results in small energy transmission. To compare the sensitivity of various sensing materials with no influence caused by the amount of coating, normalization is a necessary step. The frequency shift of a surface-acoustic-wave sensor is assumed to conform to Equation (1) for an acoustically thin and perfectly elastic thin film, *k*_1_ and *k*_2_ are piezoelectric material parameters, *f*_0_ is the central frequency of the SAW device, *h* is the thickness of the sensing film, *ρ* is the density of the sensing film, *m* is the mass of adsorbed molecules, and A is the coated area [32]. Equation (2) is derived from Equation (1) because *m*_coating_ and △*f*_coating_ are constant after coating. *m*_gas_ is linear with △*f*_gas_, which means that the frequency shift caused from an inconsistent amount of coating can be calibrated on dividing by △*f*_coating_. In this research, the normalized frequency shift is defined as a ratio of the frequency shift caused by detection and coating (△*f*_gas_/△*f*_coating_) [16].
(1)$$△f=\;\left(k1\;+k2\right)f_{0}{}^{2}h\;\rho =\;\left(k1+\;k2\right)f_{0}{}^{2}\;m/A$$
(2)$$△f_{\mathrm{gas}}/△f_{\mathrm{coating}}=m_{\mathrm{gas}}/m_{\mathrm{coating}}$$

This frequency shift caused on coating is shown in Table 1. In the non-polymer-based sensing materials, it can observe that, when the chemical oxidation is stronger, the material surface is more hydrophilic because of more oxygen-containing functional groups introduced. The deposited amount after spin coating (1500 rpm, 30 s) remaining in the sensing area (gold) is greater, which results in a large frequency shift. The HMC has a hydrophobic property. Therefore, under the same conditions of spin, the coating remains less than O-HMC. The original frequency shift and normalized frequency shift with 3-EP at varied concentrations is shown in Figure 6. Both types of sensing materials present satisfactory linearity, and the sensitivity (defined as the slopes of the regression line, △*f*_gas_/sample concentration) of PAA-HMC and O-HMC are 37.8 and 51.1 Hz/ppm. The detection limit (LOD) of the sensor is less than 1 ppm. Furthermore, it is insufficiently accurate to analyze the sensitivity of sensing materials of the two types from the original frequency shift because the deposited O-HMC is much less than PAA-HMC. The frequency shifts of the coatings (Δ*f*_coating_) were 60 kHz for O-HMC and 120 kHz for PAA-HMC. After normalization, it shows a larger difference and the sensitivity can be more clearly compared for the two sensing materials. The slope of the regression line for O-HMC is about 2.7 times that for PAA-HMC.

### 3.4. Detection of Cigarette Smoke

Besides the detection of pure 3-ethenylpyridine, cigarette smoke is also detected with the same system and same sensing material. Unlike the pure compound of 3-EP, the cigarette smoke is a complicated mixture, including toxic volatile organic compounds (VOC) of many kinds such as benzene, toluene, formaldehyde, phenol, and sticky tar, which is produced by incomplete combustion. Figure 7a is the frequency response of cigarette smoke (burning 1 cigarette) detected with O-HMC, but the sample of cigarette smoke was collected without a filter. It shows that the frequency response continuously decreases when the cigarette smoke flows into the micro-chamber. The adsorption cannot achieve a dynamic balance and desorption cannot fully return to the original resonant frequency. This abnormal phenomenon of detection was likely due to the surface adhesion of tar. When the cigarette smoke sample was collected without a filter, sticky tar and particulate matter (PM) might cover the detection area, even sticking to the porous structure of the sensing material, which makes permanent mass loading and causes incomplete desorption.

Figure 7b is the frequency response for detecting cigarette smoke with O-HMC for five successive cycles. The sample of cigarette smoke (1 cigarette burning) is collected with a 1-μm filter. The SAW frequency decreased immediately when the cigarette smoke was introduced. Compared to the sample of cigarette smoke without a filter, the filtered cigarette smoke sample took about 4 min to complete the adsorption and 15 min to complete the desorption and to recover to the original frequency. The average frequency shift was about 4200 Hz. After filtering, it can also detect repeatedly like the detection of pure 3-ethenylpyridine, but unfiltered tar still slightly influences the sensor performance in repeatability.

### 3.5. Selectivity of an Oxidized Hollow Mesoporous Carbon Nanospheres (O-HMC)-Coated SAW Sensor

When a cigarette burns, many chemical compounds are released into the air such as carbon oxide, carbon dioxide, methane, furans, PAH, nicotine, and 3-EP [4,13,33]. Because the properties of some compounds in cigarette smoke are similar to those of the cigarette marker and are also adsorbed on the sensing material, it is difficult to conclude that the total frequency shift in the detection of cigarette smoke is contributed from the cigarette marker. To determine the influence of non-cigarette- related substances, a selectivity test is necessary. Figure 8 shows a comparison of sensitivity of the SAW sensor coated with an O-HMC-sensitive material towards CO, CO_2_, CH_4_, and 3-EP (SHS marker), which indicates a high selectivity to the cigarette smoke marker because of the high affinity of 3-EP to O-HMC. These results indicate that the frequency shift due to CO and CO_2_ could be negligible during the detection of cigarette smoke, but CH_4_ shows a larger adsorption to O-HMC as the micro-pores and meso-pores in the carbon of diameter less than 2 nm has an effective adsorption capacity for methane [34]. As the SAW sensor performance is also seriously affected by moisture, the frequency shift at varied relative humidity (RH) is also measured to calibrate the influence of water molecules. Overall, the selectivity of O-HMC toward cigarette smoke is adequate to apply the SAW sensor and detection under real conditions.

## 4. Conclusions

Statistical data indicate that persons exposed to second-hand smoke (SHS) have a greater risk of lung cancer and coronary heart disease. The surface-acoustic-wave (SAW) gas sensor coated with oxidized hollow mesoporous carbon nanospheres (O-HMC) detects the second-hand smoke marker (3-ethenylpyridine) and cigarette smoke. This non-polymer sensitive coating is made through treatment with nitric acid and has many carboxyl groups to bond with the tobacco marker. The O-HMC is more sensitive than PAA-HMC and can avoid the shortcomings of a polymer-based sensing material. It shows satisfactory selectivity to CO, CO_2_, CH_4_, and 3-EP. The large specific surface area caused by the porous structure and the self-designed micro-chamber lead to rapid detection at a small flow rate. Besides the detection of the pure compound, the SAW sensor detects cigarette smoke repeatedly with a filter of suitable size to remove excess tar and particles, which have great potential as a real-time smoke detector. This SAW sensor is an important distinction from passive sampling and time-consuming measurements of ambient nicotine. It allows for a demonstration of the concentration changes in the air cigarette marker from a specific smoking condition and provides the immediate protection from tobacco hazards.

## Figures and Tables

**Figure 1 micromachines-10-00276-f001:**
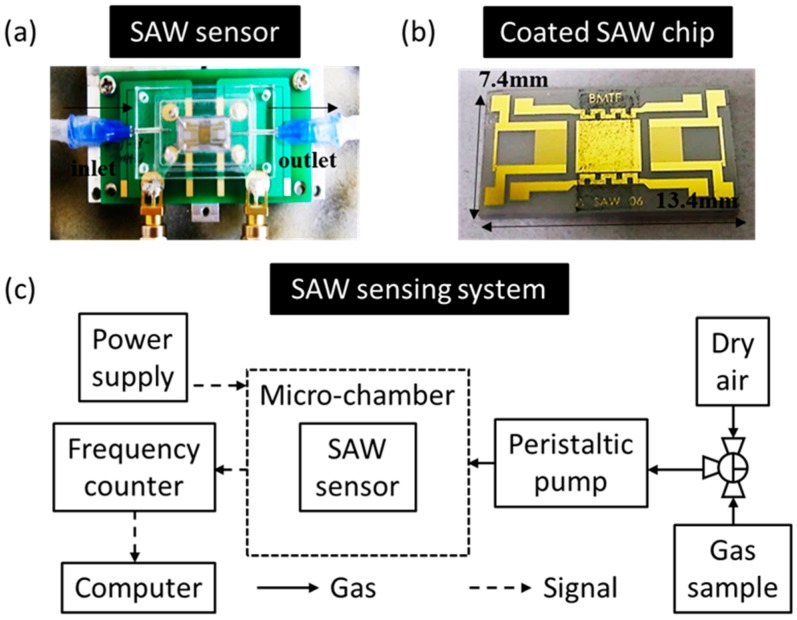
(**a**) Image of surface-acoustic-wave (SAW) sensor with a micro-chamber. (**b**) Spin-coated SAW chip with oxidized hollow mesoporous carbon nanospheres (O-HMC). (**c**) Schematic diagram of SAW sensing system.

**Figure 2 micromachines-10-00276-f002:**
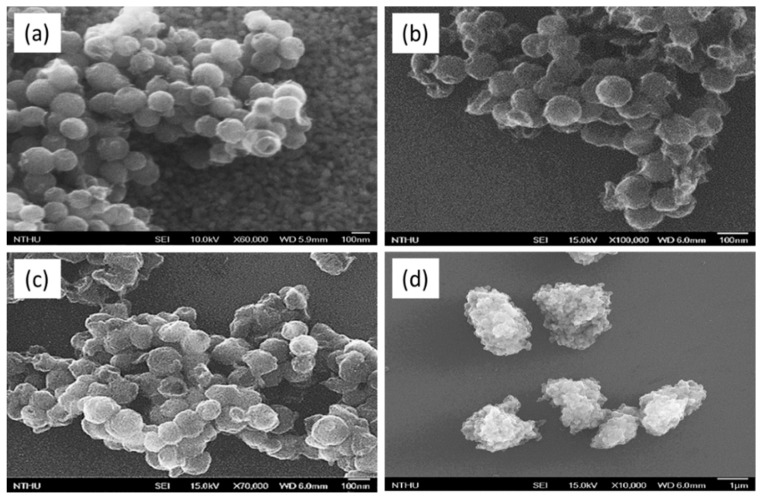
SEM images. (**a**) Hollow mesoporous carbon nanospheres (HMC). (**b**) O-HMC modified at 80 °C and 15 h. (**c**) O-HMC modified at 110 °C and 24 h. (**d**) O-HMC modified at 110 °C and 24 h on a small scale.

**Figure 3 micromachines-10-00276-f003:**
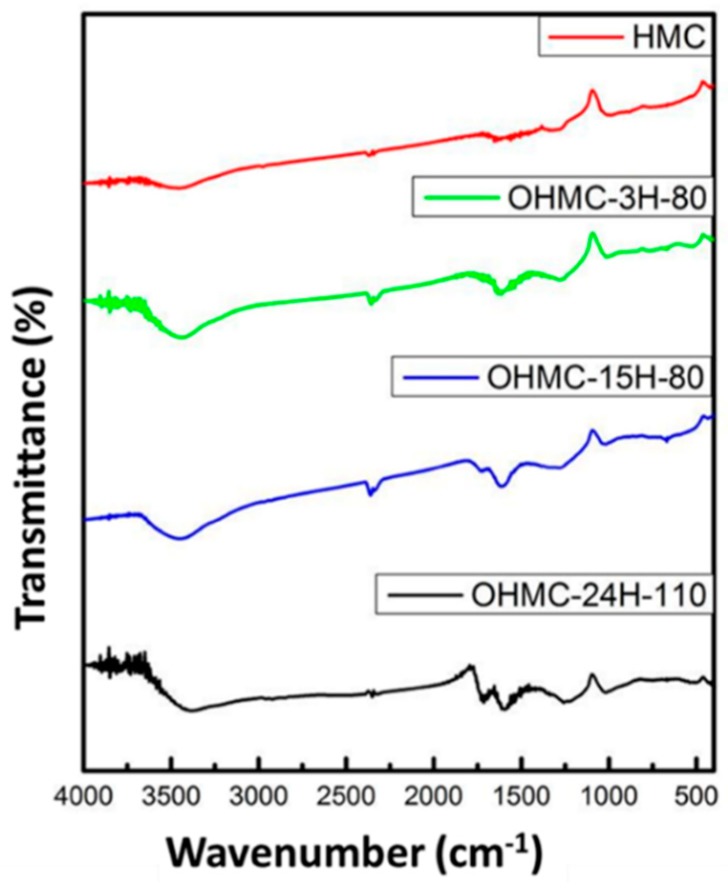
Infrared (IR) spectra of HMC and O-HMC with modified parameters.

**Figure 4 micromachines-10-00276-f004:**
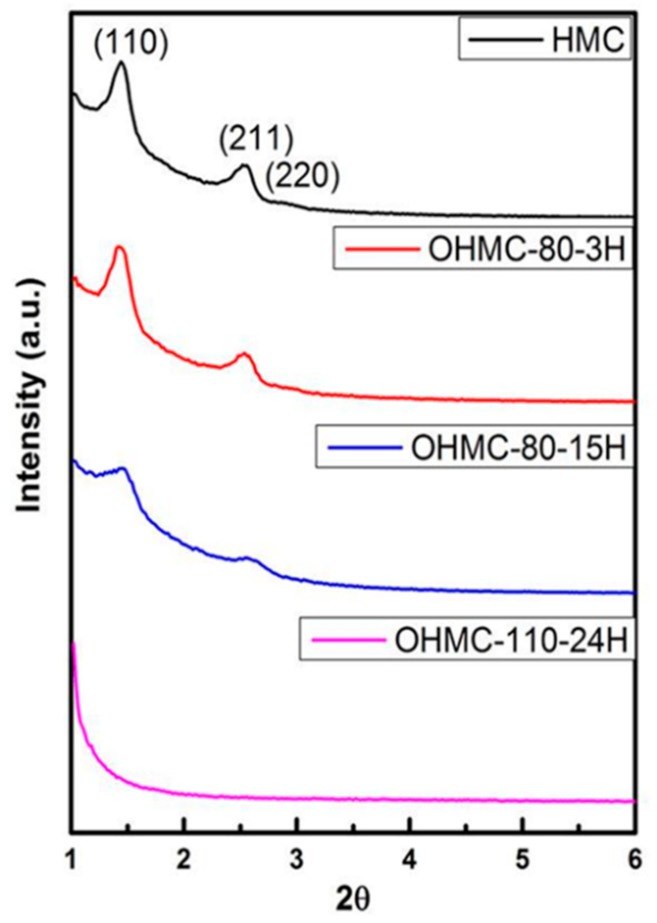
XRD of HMC and O-HMC with modified parameters.

**Figure 5 micromachines-10-00276-f005:**
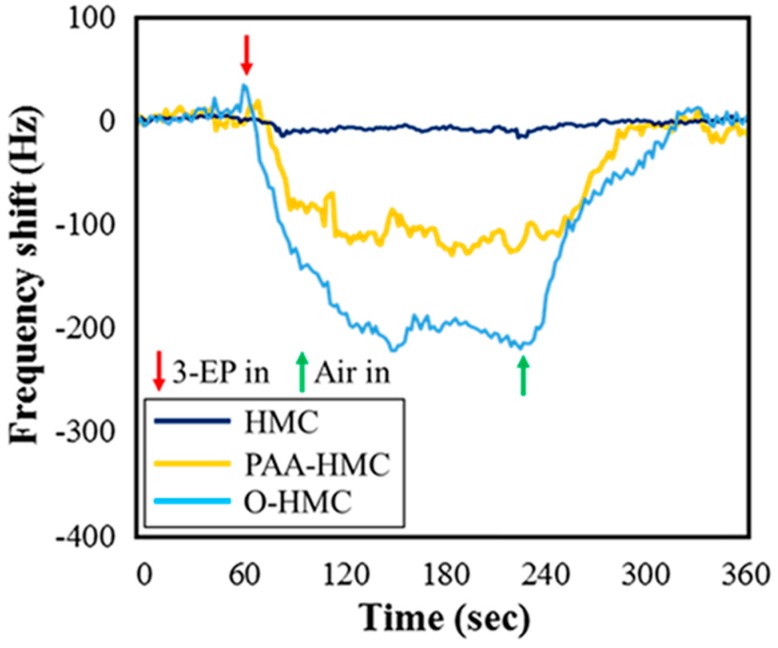
Frequency response of sensing materials for detection of 3-EP (3 ppm).

**Figure 6 micromachines-10-00276-f006:**
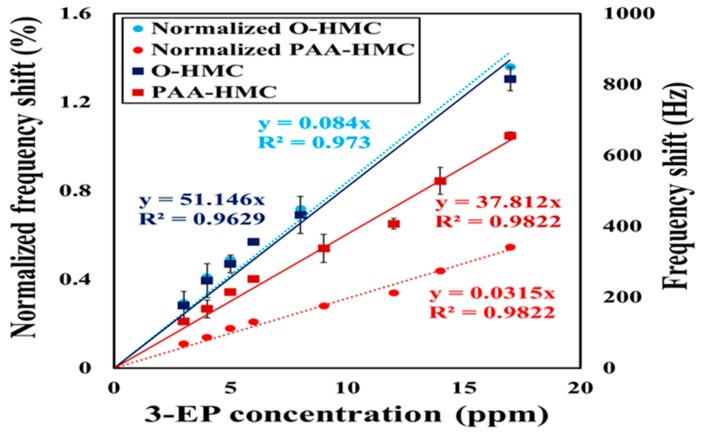
Frequency shift (deep blue and deep red) and normalized frequency shift (blue and red) of sensing materials O-HMC, polyacrylic acid, and hollow mesoporous carbon nanospheres (PAA-HMC).

**Figure 7 micromachines-10-00276-f007:**
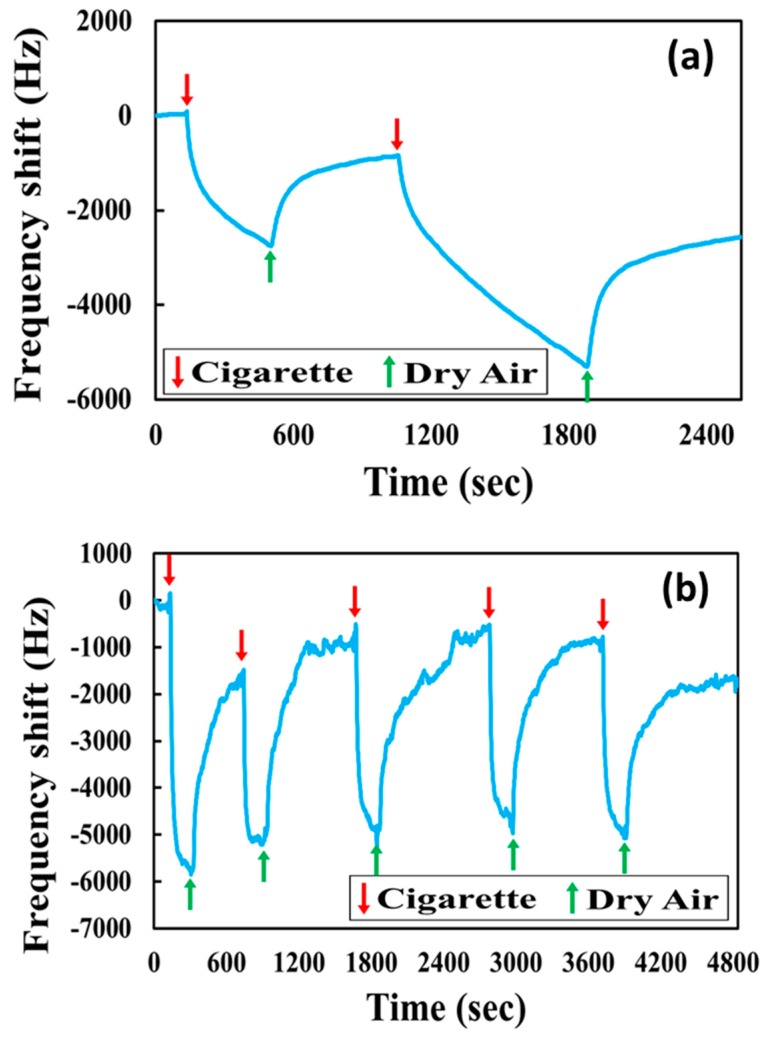
(**a**) Frequency response of the O-HMC-coated SAW sensor for detection of cigarette smoke without a filter. (**b**) Frequency response of the O-HMC-coated SAW sensor for detection of cigarette smoke with a 1-μm filter.

**Figure 8 micromachines-10-00276-f008:**
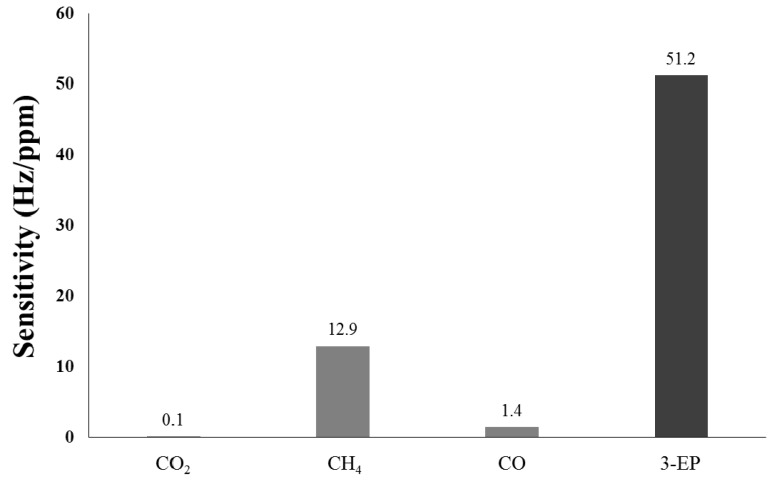
Selectivity of SAW sensor coated with O-HMC towards CO_2_ (1.27%), CH_4_ (150 ppm), CO (350 ppm), and 3-EP (standard cigarette marker, 17 ppm).

**Table 1 micromachines-10-00276-t001:** Surface-acoustic-wave (SAW) Resonant frequency before and after coating.

Sensing Materials	Before Coated (MHz)	After Coated (MHz)	Coated Frequency Shift (Hz)
Polymer based			
PAA-HMC	114.71	114.59	120,000
Non-polymer based			
O-HMC-110 °C-24 h	114.98	114.88	100,000
O-HMC-80 °C-15 h	114.12	114.07	50,000
O-HMC-80 °C-3 h	114.16	114.11	50,000
HMC	114.16	114.13	30,000

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
