# Peer review of "Detection of Cigarette Smoke Using a Surface-Acoustic-Wave Gas Sensor with Non-Polymer-Based Oxidized Hollow Mesoporous Carbon Nanospheres"

_micromachines, 2019, doi:10.3390/mi10040276_

Round 1
Reviewer 1 Report
Manuscript reports a development of SAW sensors covered with non-polymer-based oxidized hollow mesoporous carbon nano spheres for detection of several compounds in cigarette smoke. Manuscript is well motivated, contains representative introduction and literature preview, the conclusions are supported by results. Authors report the morphology studies of developed coatings, some selectivity tests and discuss the importance of SAW sensor normalization. The tests on real cigarette smoke detection were also done. I would suggest the manuscript for publication in Micromachines after minor revision: the comparison of sensor response toward smoke of different type (producers) cigarettes, as far as single measurement set including both cigarette smoke response and other different air components (try to perform measurements reported in Fig.7 and Fig 8 together). Moreover, more attention should be given to sensor reproducibility and lifetime tests.
Reviewer 2 Report
The paper “Detection of Cigarette Smoke Using a Surface acoustic-wave Gas Sensor with Non-polymer-based Oxidized Hollow Mesoporous Carbon Nanospheres” by Chi-Yung Cheng, Shih-Shien Huang, Chia-Min Yang, Kea-Tiong Tang, and Da-Jeng Yao describes an innovative topic. SAWs were combined with nano materials as coatings. These sensitive layers were characterized by spectroscopic methods. LiNbO3 was chosen as piezoelectric material for the design of the transducer. The materials quartz, LiTaO3 and LiNbO3 are widely used for the design of SAWs. The LiNbO3 material is especially applied for broadband filters and other high frequency purposes. The extraordinary property of LiNbO3 is its large electromechanical coupling coefficient. In spite of this fact it is only scarcely used for chemical sensors since they show an extremely high temperature dependence. Thus, an dual line arrangement must designed to compensate for temperature dependencies. The paper should include some measurements as function of temperature.
The following paper concerning SAWs should be cited: Surface AcousticWave (SAW) for Chemical Sensing Applications of Recognition Layers; Sensors 2017, 17, 2716; doi:10.3390/s17122716.
